## COMMENT

# Gender disparities in altmetric attention scores for cardiovascular research

Marc J. Lerchenmueller[1,2,5], Leo Schmallenbach [1,5], Maximilian Bley [3] &
Carolin Lerchenmüller [3,4 ✉]

Analysis of the association between Altmetric scores and the gender of first and last authors of articles published in top cardiovascular research journals shows that women receive on average less attention for their work.

Gender diversity at the workplace improves innovation, productivity, and profitability. In the life sciences, gender diversity ultimately benefits research and patient care[1–4]. Although the share of women in the life sciences has been increasing, the transition into senior roles stagnates and a relevant gender gap remains[5]. While awareness is rising and progress toward gender equity has been made, women still face inequities impacting their career advancement. Especially for academic careers, research funding, publications, and citations are predictive of academic rank, however, previous research has shown gender-based disparities for all of these[5].

Among the correlates of academic career progress of men and women, gender differences in citations have received increased scholarly attention[6,7]. Citations serve as the institutionalized metric for the impact of academic work and play a central role in academic advancement, from hiring to promotion[8]. Prior research has shown that women receive fewer citations than men, even for work of comparable quality[9]. As differences in the quality of work cannot explain gendered citation differentials, the contributing factors to women receiving fewer citations than men remain to be investigated.

Attention to research articles is a theoretical antecedent to citations since scholars can only cite articles they are aware of. Attention to research may thus serve as an early indicator of citation potential. Any gender difference in attention to research may conceivably perpetuate citation differentials between women and men. At the same time, differing visibility of men and women researchers or different engagement in promotion activities, e.g., due to time or budget constraints, may give rise to a gender attention gap.

In this study, we analyze potential gender disparities in the attention to research, subject to holding the content and quality of the underlying research constant to the degree feasible. To that end, we focus on the top field journals of a medical specialty and select cardiology and cardiovascular research as our empirical setting. Past research has documented a general association between attention and citations in this specialty[10], that we can build upon to analyze potential gender differences. Furthermore, past research has identified cardiology as a subfield of life science research where women are particularly challenged in their career advancement[11]. Gender differences in attention to research may be an early contributor to concomitant citation differentials and, ultimately, advancement in the field.

Investigating more than 6000 articles in the top five cardiology and cardiovascular field journals (2015–2021) with corresponding article-level Altmetric Attention Scores (ASS), we find

[1] Department of Organization and Innovation, University of Mannheim, Mannheim, Germany. [2] Leibniz Center for European Economic Research, Mannheim, Germany. [3] Department of Cardiology, Angiology, Pulmonology, University Hospital Heidelberg, Heidelberg, Germany. [4] German Center for Cardiovascular Research (DZHK), Partner Site Heidelberg/Mannheim, Heidelberg, Germany. [5] These authors contributed equally: Marc J. Lerchenmueller, Leo Schmallenbach. ✉email: carolin.lerchenmueller@med.uni-heidelberg.de

that women first and last authors receive significantly less attention for their research and that lower attention scores correlate with fewer downstream citations. Importantly, we show that women authors receive up to 20% fewer citations than men to articles that receive the same amount of attention. This gendered correlation of attention to citations afflicts articles in the top quartile of the ASS distribution and might markedly impact scientific discourses.

## Results

**Gender differences in overall attention and sources of attention**. Our sample consists of 6068 and 6181 articles for which we can designate the gender of the first or last author, respectively. In this sample, we first investigate the relative difference in the overall AAS as well as its constituent score components between women and men serving in the first author position (Fig. 1a) and the last author position (Fig. 1b). First authorship by women relative to men is associated with an 11.2% (95%CI -18.5%; -3.9%, $p = 0.003$) lower AAS. For the last authors, the discount is slightly smaller with women having an 8.7% (95% CI -17.1%; -0.1%, $p = 0.044$) lower AAS than men on average. We next disaggregate our data by attention source, including News mentions, Blogs, Wikipedia entries, Policy documents, and Twitter mentions. We also stratify the Twitter data by the demographics of underlying user groups. We find that the reduced attention to research by women first and last authors mostly stems from the social media platform Twitter (Fig. 1a, b). The analysis of different Twitter user groups further reveals that the discount is driven both by members of the public and members of the scientific community. Lastly, it is worth mentioning that both women as first and last authors tend to be less represented in their research in policy documents, albeit this trend does not statistically distinguish from a null effect.

**Gender differences in citations**. We next investigate whether this gender difference in attention is also correlated with gender differences in citations. In our sample, we find that women first authors receive 5.6% (95%CI −10.2%; −1.1%, $p = 0.014$) and women last authors 6.6% (95%CI −11.8%; −1.4%, $p = 0.014$) fewer citations, on average (Fig. 2a). Having established this gender difference in citations, we then investigate the conditional correlation between AAS and citations stratified by author gender. Figure 2b shows that women first authors receive fewer citations to research that attracted the same level of attention as men. This gender difference is statistically significant for articles

in the top 25% of the AAS distribution (i.e., log(AAS + 1) of 4.5/ raw AAS of 90). On average, this gender difference amounts to a 20.0% lower correlational conversion rate (95%CI, −30.0%; −10.0%, $p < 0.001$) from AAS to citations for women versus men first authors. We observe a similar trend for last authors (−11.6%, 95%CI −24.6%; 1.6%, $p = 0.086$). This implies that women first (last) authors receive, on average, fewer citations than men first (last) authors for research that attracts a lot of attention and citations to begin with and might thus be more likely to spur public debate and scientific progress.

**Authorship gender composition and attention to research**. An additional analysis of the combination of first and last author gender (Fig. 3) further reveals that accounting for confounders, articles authored by both women as first and last authors receive the lowest AAS (Fig. 3a), the lowest number of citations (Fig. 3b), and lowest correlational conversion rate between AAS and citations (Fig. 3c). Publications authored by men first and last authors receive the highest values on these outcome measures, whereas mixed gender author teams range in between. This gradient in effect sizes supports the overall finding that gender matters for attention to research, potentially putting women at a relative disadvantage. This analysis is limited to 5678 articles with at least two authors and where both first and last author gender can be designated.

## Discussion

To our knowledge, this study is the first to document the presence and size of a gender bias in attention to cardiovascular research and to decompose this gender attention gap by source. Additionally, we create a correlational link between attention and citations, which also appears to be gendered. Earlier investigations of the correlation between AAS and subsequent citations of research showed only a weak correlation in top tier journals in cardiology and cardiovascular research[12,13]. However, the rising use of social media to disseminate science likely increases the importance of the AAS for forward citations, as reflected in more recent research[14–16]. Our study extends these findings by estimating the conditional correlation between AAS and citations and, importantly, stratifying the conditional correlation by gender.

Together, the identified correlational cascade from attention to citation differentials between women and men may pinpoint a root cause for gendered recognition of research contributions. The scientific community has gained a robust understanding of

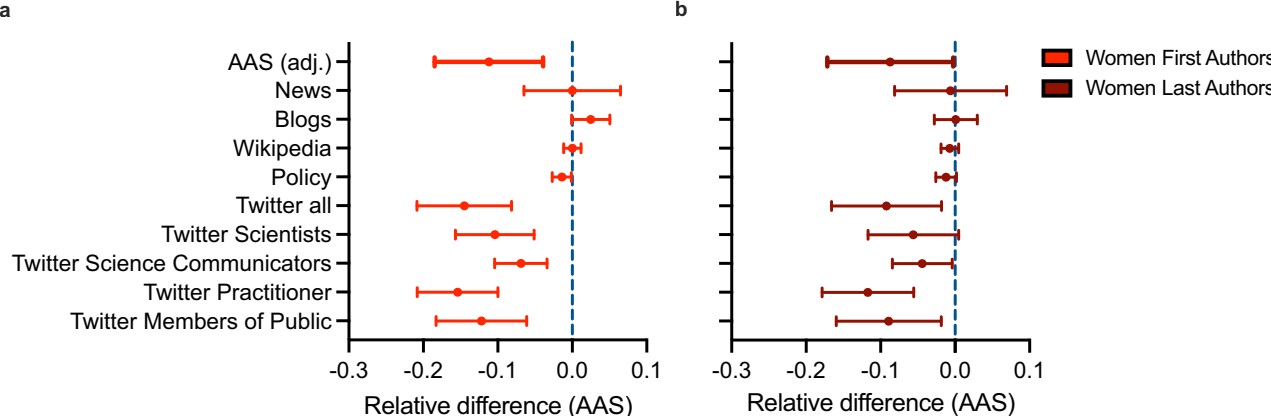

**Fig. 1 Gender differences in overall attention and sources of attention.** Relative differences—women (red lines) versus men (blue dotted line)—in Altmetric Attention Score (AAS) and its constituent score components (e.g., News, Blogs) split by **a** first author (bright red line, $n = 6068$) and **b** last author (dark red line, $n = 6181$). Error bars represent 95% confidence intervals.

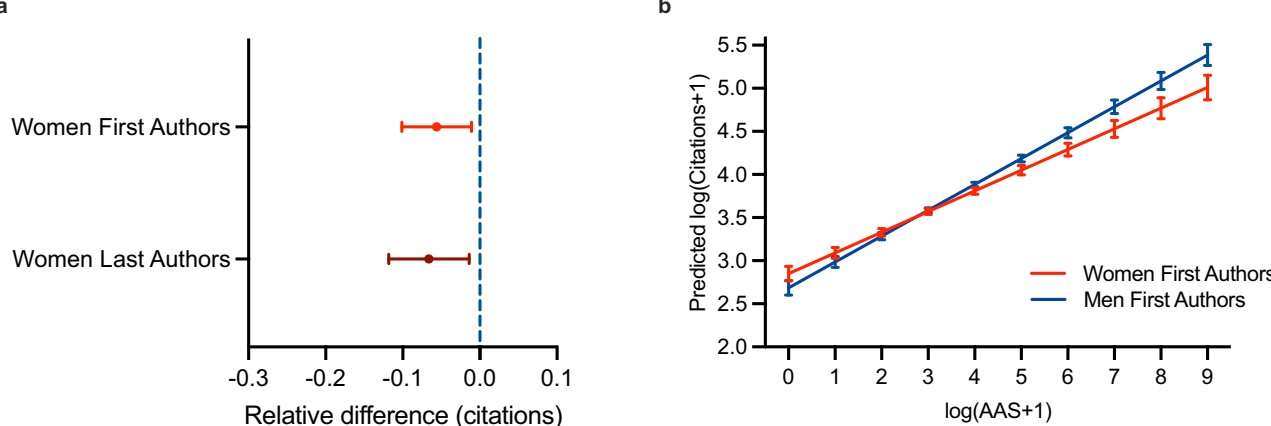

**Fig. 2 Gender differences in citations and correlational translation of attention to citations. a** Gender differences in citations for women first authors (bright red line n = 6068) and women last authors (dark red line, n = 6181) relative to men in respective authorship positions (blue dotted line). **b** Predicted marginal effects of the interaction effect of first author gender and AAS with respect to predicted citations (n = 6068). Overall, data is based on OLS regressions, controlling for articles' publication month and year, the publishing journal, the number of co-authors, and whether the article was COVID19 related. Error bars represent 95% confidence intervals.

**Fig. 3 Authorship gender composition and attention to research. a** Differences in AAS by the gender combination of first (FA) and last authors (LA) (women FA&LA = red line, mixed gender FA&LA = green line, n = 5678) relative to both first and last author being men (dotted blue line). **b** Differences in citations by gender combination of first (FA) and last authors (LA) (women FA&LA = red line, mixed gender FA&LA = green line) relative to both first and last author being men (dotted blue line, n = 5678). **c** Predicted marginal effects of the interaction effect of gender combination of first and last authors with respect to predicted citations (n = 5678). Error bars represent 95% confidence intervals.

gender differences in recognition of work at later stages of the academic lifecycle, like gender differences in promotion, the awarding of scientific prizes, and the representation of men and women in esteemed scientific societies[17–20]. Our results, meanwhile, shed light on a possible, much earlier starting point of gender differences in scientific recognition—the attention paid to a research article—that may conceivably contribute to downward spirals of recognition as the academic lifecycle continues.

The presented findings hold several implications for science policy and practice. As the importance of broader attention metrics becomes increasingly recognized by the institutions of science[21], from funders to universities, it seems important to raise awareness of potential differences in attention to research and likely consequences for more established metrics like citations. Awareness is the first step for researchers to make conscious decisions about how to represent their work to the scientific community and the broader public[22]. Meanwhile, institutions may need to consider how to formalize financial and non-financial (e.g., media consulting) support for their research staff when it comes to professionally disseminating research findings and should consider equity measures. Our research indicates that gender equity considerations are likely important for designing such support structures, whereas similar needs might exist at a more intersectional basis (race, ethnicity etc.)[23]. For the scientific research enterprise at large, attention allocation will likely play a central role in determining what science impacts cumulative knowledge building. As scientific production proliferates, scientists become ever more constrained in the content they can absorb and use in their own research[24]. The process of attention allocation to research may, therefore, become more active field of research to promote evidence-based science policy and practice.

In that vein, our study also highlights avenues for future research. For example, we still lack a clear understanding of the antecedents of the observed gender discount in attention to cardiovascular research as well as potential consequences, for example, gendered influences of research on interventions and public policies. As a starting point, analyses of gender differences across scientific disciplines may help to better understand root causes and, for example, test whether a higher representation of women in a discipline helps to level the playing field. While not the explicit focus of this current study, past research has suggested that homophily influences citations (i.e., men tend to cite men and women tend to cite women), although empirical findings on the conjecture have been mixed[25,26]. Given that attention scores appear to correlate with citations, one might suspect that gender differences in attention might be more pronounced in fields like cardiology with relatively fewer women scientists and thus precede any citation differentials. Another area for future research relates to gender differences in attention and the subsequent impact of research on policy. Although our limited sample precludes drawing definite conclusions, the trend of women's research being cited at a lower rate than men's in policy documents warrants further attention. Prior research has, for example, indicated that women researchers were less heard than expected compared to male peers, accounting for scientific expertise, during the formation of the societal response to the COVID19-crisis[27,28]. We therefore submit that further analyses of gender differences with respect to the impact of scientific work beyond citations, like informed policy making, is needed.

Limitations of our study include potential bias due to omitted variables, possible gender misclassification in a large set of authors, and more granular time variance in AAS and citations (beyond the year-month-level we account for).

In conclusion, this study shows that women receive less attention for their work (lower AAS) than men in the top cardiology and cardiovascular research journals. Gender differences in AAS were mostly driven by social media posts on Twitter and correlate with fewer citations for women scholars, both in absolute and relative terms. Causes of this gap can only be speculated at this point. Assuming homogeneous quality of the underlying research given the selected sample, contributing factors may include gender bias among peers and other social media users, but also differential self-promotion patterns[22]. This supports recent initiatives to increase the visibility of women academics, but also calls to encourage women to engage in social media platforms and actively disseminate their work. That said, promotional approaches seem constrained by the fact that women receive differential recognition in form of citations for work that garners equal attention.

## Methods

**Sample creation**. To analyze gender differences in attention to cardiology and cardiovascular research, we first identified the top five field journals per the Clarivate Journal Citation Report (2020). Using the unique International Standard Serial Numbers (journal ISSNs) as identifiers, we retrieved original research articles from the central bibliometric database in the life sciences, the *PubMed* database administered by the U.S. National Library of Medicine. We focused on articles published between 2015/01/01 and 2021/12/31 (or since the first issue in the case of JAMA Cardiology), a time window for which the relatively new AAS reliably tracks attention to articles. We retrieved the AAS and its constituent score components from the Altmetric application programming interface (API).

We focus on the first and last authors because of a long-standing authorship norm in the life sciences. The first author of life science articles usually represents the junior author who executed the research, while the last author is generally the senior author who often conceives of and funds the research. To designate the probable gender of thousands of these authors in our dataset, we use the genderize.io database that draws on several sources, like Social Security Administration records and social media profiles, to assign a probability that a given forename is more likely held by men or women[11,22,29]. The average probability for correct gender designation in our dataset was 96.3% for first authors and 97.0% for last authors. Within our sample of high-impact Cardiology and Cardiovascular research, we found fewer manuscripts authored by women as first (31.74%) or last (19.97%) authors. Our final sample consists of 6068 articles for which we could designate the first author gender and 6181 articles for which we could designate the last author gender.

**Statistics and reproducibility**. We use ordinary least squares regressions to analyze the relationship between our key independent variable, lead author gender, and our dependent variables of attention and citations. We follow prior research in log-converting the AAS and its components to account for the scores' skewness[16]. We control for an encompassing set of confounders, including the timing of publication at the year-month-level (attention accrual may vary with time), research quality (using journal dummy variables), and social network exposure (proxied by the number of coauthors). We also include a dummy variable that controls for COVID-19-related articles since COVID-19 research received heightened attention. COVID-19-focused papers accounted for 1.6% of our sample, and we do not find gender differences in AAS in this subset. We limit our choice of control variables to factors that may be spuriously related to both gender and AAS but are unlikely to mechanistically explain their relationship. We use STATA software (Version 17) for regression analyses and GraphPad PRISM software (Version 8) for the creation of Figures.

**Reporting summary**. Further information on research design is available in the Nature Portfolio Reporting Summary linked to this article.

## Data availability

The data underlying our analysis is available at the Harvard Dataverse https://doi.org/10.7910/DVN/OAYQA8.

## Code availability

The code for replicating our analysis in Stata 17 is available at the Harvard Dataverse https://doi.org/10.7910/DVN/OAYQA8.

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

## Acknowledgements

This research has received funding from the Dr. Hans Riegel Foundation and the Deutsche Forschungsgemeinschaft within the funding programme 'Open Access Publikationskosten' as well as by Heidelberg University. LS is supported by the Joachim Herz Foundation.

## Author contributions

M.J.L., L.S., and C.L. devised the original idea. M.J.L. and L.S. assembled and analyzed the data. M.B. reviewed relevant literature. M.J.L., L.S., and C.L. wrote the manuscript, M.B. edited the manuscript.

## Funding

## Competing interests

M.J.L. is a co-founder and shareholder of AaviGen GmbH, a cardiovascular gene therapy company, which is unrelated to this article. The remaining authors have no competing interests.
