## [Peer review file · Communications Biology]

Reviewers' comments:

Reviewer #1 (Remarks to the Author):

This study showed that women receive less attention for their work (lower altmetric adjusted score) than men in the top cardiology and cardiovascular research journals and these gender differences were mostly driven by social media posts on Twitter and correlate with fewer citations for women scholars, both in absolute and relative terms. Furthermore, for equivalent attention on social media, women's research attracted fewer citations than men's research. Overall, this is an excellent manuscript. The analysis is extremely relevant considering the rising importance of social media in science dissemination. The methodology is clearly described and robust. The findings compellingly illustrate women's disadvantage in academia and research, thus extending evidence already available on women's underrepresentation among authors, reviewers and editors of scientific journals. It is a pity that the brief format precludes a more comprehensive discussion of the findings and their implication for advocacy toward gender equity in science. Congratulations to the authors on this very well-written and timely manuscript.

I have only a few comments/suggestions:

1. It's interesting that the lines cross in Figure 2b – could the authors possibly comment on this?
2. Could the authors comments on the borderline under-representation of women in policy mentions? This is extremely important as policy informs public health practice and medicine and has a major impact on population health. Gender bias among policy makers is a major issue and may be even more strongly associated with social media than citations in scientific manuscripts because policy makers may be more likely to be influenced by social media. I understand numbers are small and hence a formal analysis would likely be underpowered.
3. Line 51 – I suggest changing the sentence to “has been increasing”, etc.

Reviewer #2 (Remarks to the Author):

Thank you for the opportunity to review this manuscript. This brief report evaluates the association between Altmetric scores and the genders of first and last authors. The sample is limited to articles published in high impact cardiovascular journals.

Major comments:

Prior studies have shown (1) the relationship between authors' genders and Altmetric scores (ref#13); (2) that women have smaller audiences on academic Twitter (Zhu et al, JAMA Internal Medicine, 2019); and (3) that Twitter amplification is associated with more citations of academic work (ref #11). The novel point in this work is that the relationship between Altmetric scores and citations is attenuated by gender, and that conditional on having the same Altmetric score, women are still cited less often (Figure 2B).

This is an interesting contribution and highlights how social media may not serve to equalize the playing field in academia, but rather may perpetuate similar network effects that have led to promotion and amplification differences that preceded the Altmetric era. Reframing the manuscript around this major point would be useful.

It is also interesting that the relationship between AAS scores and citations is flipped for articles at the lower end of the AAS distribution. Around a value of 3 on the x-axis of Figure 2B, there appears to be a "tipping point" at which the citation returns to additional altmetric attention are diminished among women. It would be helpful to know what score that translates to in absolute terms, and where that score falls in terms of the distribution of AAS scores among academic publications more broadly. The narrow error term at that point seems to suggest that that is where most articles in the sample tend

to fall in terms of their AAS scores, in which case, does the conditional difference only become a problem for heavily cited articles?

Minor points

-Why these patterns would differ in the field of cardiology is not clearly motivated. Is it because women are less represented in cardiology across clinical fields? Why not surgery and cardiology? Would you expect the opposite patterns in obstetrics/gynecology or pediatrics?

-I am unable to view details in Dataverse as it states "to be completed". I was primarily interested in evaluating the Covid articles separately, as prior work has shown that Covid exacerbated many issues related to the advancement of women in academic medicine.

Reviewer #3 (Remarks to the Author):

"Gender Disparities in the Attention to Cardiovascular Research," by Marc J. Lerchenmueller and colleagues, evaluates the relationship between the Altmetric Attention Score (AAS) and citations among male and female first and last authors of all research manuscripts published in five leading cardiovascular care journals (Circulation, JACC, JAMA-Cardiology, European Heart Journal, and Circulation-Research) between 2015-2021. The authors find that articles with female first and/or last authors have lower AAS scores than those with male first and/or last authors. In addition, even after adjusting for multiple potential confounders, articles with first and last authors who are women are less likely to be cited than authors with similar AAS scores published by men. They also find that the differences in AAS scores between articles with female and male authorship appear to relate to differences in how these articles are promoted on Twitter (and particularly how scientists and the lay public comment on them). This last finding is the most interesting and potentially relevant aspect of the study. If accurate, this finding has important implications for how research is disseminated and gender gaps in dissemination/promotion strategy. However, before "buying into" this finding, I would want to see the authors do a number of additional analyses to enrich the analysis and address potential unmeasured confounders. I have several thoughts and recommendations for how to do this.

1. Controlling for unmeasured confounders:

a. Do the authors control for year of publication in addition to month? If yes, please state as much. If not, year of publication should be controlled for, because if men had more publications earlier in the study period then these publications have had a longer period of time to be cited. clear that they also control for year of publication.

b. Several other factors that the authors didn't evaluate may influence AAS, and I'd like to see them try to control for these confounders. A few examples:

i. Institutions vary in their support for promoting research products. Some hospitals, health systems, and medical schools are great at it; others put very little effort into this. If men were more likely to be employed by institutions with savvy marketing/promotion, then this could help account for AAS differentials.

ii. Similarly, we already know that there is a sex difference in research funding rates in academic medicine, and in numbers of publications. To the extent that these factors influence the success of efforts to promote research (because people with research funding can devote resources to promotion and those with more publications have more experience with promotion and are better at it), this could also be driving the outcomes observed in this study.

1. One way to try to address this would be to control for each author's unique first and last name using a fixed effects model (because the authors' names are associated with their sex). This would effectively enable you to control for all characteristics of each author (including their institution, their publications, and their research funding) without controlling for their sex.

2. I would like to see the authors assess whether having a female first AND last author, and similarly a male first AND last author, result in even lower AAS scores and citations (in case of female first and last author) and even higher AAS scores and citations (in case of papers with male first and last

authors). If sex is indeed causally linked to AAS score and citations, then we would expect that having both a first and last author who are female or male would result in a more intense effect size. Perhaps looking at adjusted AAS and adjusted citations among three classes of publications:

a. Female first and last author

b. 1 female author and 1 male author

c. Male first and male last author

3. Line 112: The authors use the term "conversion rate" which suggests a causal link between AAS and citations. As far as I'm aware, they have not proven a causal link between AAS and citations. Instead, they have identified an association. I would revise this language accordingly.

Re: COMMSBIO-22-3829-T “Gender Disparities in the Attention to Cardiovascular Research”

Point-by-Point Response to Reviewers

We would like to thank the reviewers for taking the time to thoroughly evaluate our work. We are pleased with the considerable interest in our manuscript and greatly appreciate the detailed critiques and suggestions. In response, we revised the manuscript and performed additional analyses. In particular, we have specified the choice of controls for confounders and added an analysis of the attention allocated to manuscripts contingent on the gender composition of the lead first and last authors – all female, all male, and mixed gender. Below we have pasted the Reviewers’ comments in **black**, followed by our response in **blue**. We have numbered the comments for convenience.

Reviewer #1 (Remarks to the Author):

Expertise: gender disparities + cardiovascular

This study showed that women receive less attention for their work (lower altmetric adjusted score) than men in the top cardiology and cardiovascular research journals and these gender differences were mostly driven by social media posts on Twitter and correlate with fewer citations for women scholars, both in absolute and relative terms. Furthermore, for equivalent attention on social media, women’s research attracted fewer citations than men’s research. Overall, this is an excellent manuscript. The analysis is extremely relevant considering the rising importance of social media in science dissemination. The methodology is clearly described and robust. The findings compellingly illustrate women’s disadvantage in academia and research, thus extending evidence already available on women’s underrepresentation among authors, reviewers and editors of scientific journals. It is a pity that the brief format precludes a more comprehensive discussion of the findings and their implication for advocacy toward gender equity in science. Congratulations to the authors on this very well-written and timely manuscript.

I have only a few comments/suggestions:

R1.1. It’s interesting that the lines cross in Figure 2b – could the authors possibly comment on this?

The findings shown in Figure 2b indicate that women experience a disadvantage in attention for publications that receive the most attention, i.e., for research where attention differences likely matter the most for science dissemination. The 95% confidence intervals of the predicted marginal values for the logged number of citations given an article’s attention score overlap for women and men first authors at values smaller than a logged attention score of 4.5. This indicates that the correlational conversion rate from attention score to citations does not statistically differ by gender for articles that receive relatively little attention. However, for articles that receive marked attention, i.e., that fall into the top 25th percentile of the Altmetric Attention Score (AAS) in our data, men first authors receive, on average, more citations than women first authors.

→ We included a more detailed description and interpretation of our data to the main manuscript (Results lines 175-187, Discussion lines 4-7)

R1.2. Could the authors comments on the borderline under-representation of women in policy mentions? This is extremely important as policy informs public health practice and medicine and has a major impact on population health. Gender bias among policy makers is a major issue and may be even more strongly associated with social media than citations in scientific manuscripts because policy makers may be more likely to be influenced by social media. I understand numbers are small and hence a formal analysis would likely be underpowered.

We agree that the trend towards women- relative to men-authored articles informing policy documents and guidelines raises concerns about a pluralistic representation of scientific findings. As you rightfully point out, the share of articles cited in policy documents in our sample is low (<10%), cautioning definite conclusions from the publications in our analysis. To fully test and describe the presence and size of a gender bias in policy citations, a larger sample is needed, and we encourage future research in this direction.

→ We have included this important point in our description of results (lines 159-161) as well as the discussion, particularly as a call for future research (lines 244-252).

R1.3. Line 51 – I suggest changing the sentence to “has been increasing”, etc.

→ Thank you for this suggestion, we have revised the sentence accordingly (line 100).

Reviewer #2 (Remarks to the Author):

Expertise: Gender disparities + altmetric analysis

Thank you for the opportunity to review this manuscript. This brief report evaluates the association between Altmetric scores and the genders of first and last authors. The sample is limited to articles published in high impact cardiovascular journals.

Major comments:

R2.1 Prior studies have shown (1) the relationship between authors' genders and Altmetric scores (ref#13); (2) that women have smaller audiences on academic Twitter (Zhu et al, JAMA Internal Medicine, 2019); and (3) that Twitter amplification is associated with more citations of academic work (ref #11). The novel point in this work is that the relationship between Altmetric scores and citations is attenuated by gender, and that conditional on having the same Altmetric score, women are still cited less often (Figure 2B).

This is an interesting contribution and highlights how social media may not serve to equalize the playing field in academia, but rather may perpetuate similar network effects that have led to promotion and amplification differences that preceded the Altmetric era. Reframing the manuscript around this major point would be useful.

In the revised version of the manuscript, we added an in-depth discussion and more detailed explanation of our contribution that shows women are cited less often, even when research received the same attention. We particularly emphasize the point concerning possible constraints of social media in promoting gender equity in science and call for more research in this domain. To our knowledge, our study is the first to document the presence and size of a gender discount in attention to cardiovascular research and to decompose the discount by source, e.g., mentions of articles on social media platforms in addition to the above-mentioned correlational link between citations and attention score.

→ We have included a more detailed description of this important point in our results description and discussion (lines 153ff. and 202 ff.).

R2.2 It is also interesting that the relationship between AAS scores and citations is flipped for articles at the lower end of the AAS distribution. Around a value of 3 on the x-axis of Figure 2B, there appears to be a "tipping point" at which the citation returns to additional altmetric attention are diminished among women. It would be helpful to know what score that translates

to in absolute terms, and where that score falls in terms of the distribution of AAS scores among academic publications more broadly. The narrow error term at that point seems to suggest that that is where most articles in the sample tend to fall in terms of their AAS scores, in which case, does the conditional difference only become a problem for heavily cited articles?

As you rightfully point out, the 95% confidence intervals of the predicted marginal values of the logged number of citations given an article's attention score overlap for women versus men first authors at values smaller than a logged attention score of ~4.5. This corresponds to an unlogged attention score of approximately 90 and a predicted number of citations of approximately 55. In our sample, 55 citations mark the 60th percentile of the citation distribution. In other words, the conditional difference becomes problematic for the 40% most cited articles, which arguably can be expected to be the articles that shape scientific progress and public policies.

→ We included a more detailed interpretation of the gender difference in the relationship between AAs and citations to the main manuscript (lines 175-187.).

Minor points

R2.3 Why these patterns would differ in the field of cardiology is not clearly motivated. Is it because women are less represented in cardiology across clinical fields? Why not surgery and cardiology? Would you expect the opposite patterns in obstetrics/gynecology or pediatrics?

Past research has suggested that women's representation in scientific fields influences the reception of women's work, e.g., via homophily in citation patterns (men cite men and women cite women). As such, one might suspect that if attention scores correlate with citations, gender differences in attention might be more pronounced in fields where women are less well represented and precede citation differentials. The field of cardiology has historically exhibited a low representation of women, lower than many other fields of academic medicine. Past research has further demonstrated, that despite progress in women's representation at the earliest career stages, there is a persistent and wide gender gap as careers progress (see e.g., Lerchenmüller et al. *Circulation* 2018). These considerations made cardiology a logical starting point for our investigation. Surgery might exhibit similar dynamics, yet our study precludes us from generalizing. Likewise, although the gender composition in some fields, like obstetrics/gynecology, might be more balanced, recognition on research contributions might not be unbiased either. As you point out, our findings confirm the importance of attention for citations, yet our results also show that attention is no panacea since women receive less benefit from it in terms of getting their best work recognized in the science community. While our focus on cardiology does not allow us to generalize to other fields, we suspect that the dynamic extrapolates to other fields and we call for further research testing our findings in other academic fields.

→ We have added an explanation for our motivation to focus on cardiology in the introduction (lines 127-129.) and a call for future research to investigate potential gender attention gaps in other fields in the discussion section to the revised manuscript (lines 236-244.)

R2.4 I am unable to view details in Dataverse as it states "to be completed". I was primarily interested in evaluating the Covid articles separately, as prior work has shown that Covid exacerbated many issues related to the advancement of women in academic medicine.

We apologize the data was not yet available for the review process. While we fully agree that it is interesting to evaluate COVID-related articles, those accounted for only 1.6% of our sample,

which is why we did not further elaborate on COVID-related articles in this manuscript but kept COVID as a control variable in our analyses.

- We now included a reviewer link for the data and analysis underlying our exhibits and included a data availability statement (lines 312-316): <https://www.dropbox.com/sh/vjee5z3sii9wr9a/AACip3ShFcj1kkS6syCGYae-a?dl=0>, we will upload these files to the Harvard Dataverse upon acceptance of the manuscript to enable replication and further probing of our findings by the science community.
- We have included the result that we did not find gender differences in the subset of COVID-related articles in our sample in the main text of the manuscript (lines 305-307ff.)

Reviewer #3 (Remarks to the Author):

Expertise: gender disparities + cardiovascular

“Gender Disparities in the Attention to Cardiovascular Research,” by Marc J. Lerchenmueller and colleagues, evaluates the relationship between the Altmetric Attention Score (AAS) and citations among male and female first and last authors of all research manuscripts published in five leading cardiovascular care journals (Circulation, JACC, JAMA-Cardiology, European Heart Journal, and Circulation-Research) between 2015-2021. The authors find that articles with female first and/or last authors have lower AAS scores than those with male first and/or last authors. In addition, even after adjusting for multiple potential confounders, articles with first and last authors who are women are less likely to be cited than authors with similar AAS scores published by men. They also find that the differences in AAS scores between articles with female and male authorship appear to relate to differences in how these articles are promoted on Twitter (and particularly how scientists and the lay public comment on them). This last finding is the most interesting and potentially relevant aspect of the study. If accurate, this finding has important implications for how research is disseminated and gender gaps in dissemination/promotion strategy. However, before “buying into” this finding, I would want to see the authors do a number of additional analyses to enrich the analysis and address potential unmeasured confounders. I have several thoughts and recommendations for how to do this.

Controlling for unmeasured confounders:

R3.1. Do the authors control for year of publication in addition to month? If yes, please state as much. If not, year of publication should be controlled for, because if men had more publications earlier in the study period then these publications have had a longer period of time to be cited. clear that they also control for year of publication.

We control for the interaction terms of publication year and publication month and have revised our manuscript to make this more explicit. Controlling for year is important, as you point out, to account for the accrual time of attention. Interacting month and year accounts for temporal effects at an even more granular level.

- We have specified that we control for publication time at the month-year-level (lines 302 ff).

R3.2. Several other factors that the authors didn’t evaluate may influence AAS, and I’d like to see them try to control for these confounders. A few examples:
i. Institutions vary in their support for promoting research products. Some hospitals, health systems, and medical schools are great at it; others put very little effort into this. If men were

more likely to be employed by institutions with savvy marketing/promotion, then this could help account for AAS differentials.

ii. Similarly, we already know that there is a sex difference in research funding rates in academic medicine, and in numbers of publications. To the extent that these factors influence the success of efforts to promote research (because people with research funding can devote resources to promotion and those with more publications have more experience with promotion and are better at it), this could also be driving the outcomes observed in this study. 1. One way to try to address this would be to control for each author's unique first and last name using a fixed effects model (because the authors' names are associated with their sex). This would effectively enable you to control for all characteristics of each author (including their institution, their publications, and their research funding) without controlling for their sex.

We agree that it is important to control for potential confounders that are spuriously related to both attention score and gender of the authors. Running an author fixed effects model is, unfortunately, not possible as gender is constant within authors (perfect collinearity). Consequently, a gender effect would not be estimated if author fixed effects were included. Also, fixed-effects can account for time-stationary differences across actors (even if unobserved, like innate ability), but their inclusion would not guarantee a clean identification because the scientific abilities of actors might change over time in ways meaningful to both attract attention to their research as well as e.g., the quality of their institution.

Additionally, there is an important distinction between confounders, which we would want to control for, and potential mediators that may mechanistically explain part of the observed relationship. Gendered access to funding or differences in marketing budgets and publishing experience could well be such mechanisms. The goal of our study is to descriptively establish whether women and men authored cardiovascular research receives a different amount of attention, holding the characteristics of the research (e.g., quality and timing) constant. We therefore limit our control variables to reflect article characteristics.

→ We have an explanation for the choice of control variables (lines 302-309)

While we cannot determine mechanistic explanations beyond attributions to the underlying science itself in the scope of this study, we agree that it is important to better understand the antecedents and factors contributing to the identified discount.

→ We added a corresponding call for potential future research directions (lines 233 ff.)

R3.3. I would like to see the authors assess whether having a female first AND last author, and similarly a male first AND last author, result in even lower AAS scores and citations (in case of female first and last author) and even higher AAS scores and citations (in case of papers with male first and last authors). If sex is indeed causally linked to AAS score and citations, then we would expect that having both a first and last author who are female or male would result in a more intense effect size. Perhaps looking at adjusted AAS and adjusted citations among three classes of publications:

a. Female first and last author, b. 1 female author and 1 male author, c. Male first and male last author

We agree that considering the gender of both first and last authors jointly is informative for the overall implications of our study. We have therefore conducted the analysis you suggested and find the expected gradient in effect sizes. The correlational conversion rate between attention score and citations is lowest for publications by women first and last authors and highest for publications by men in both prestigious authorship positions. Mixed gender author teams range in between (see Figure 1 below). We find the same gradient in effect sizes when testing the direct relationship between the gender of both first and last authors and the adjusted AAS (Figure 2,

Panel A) and adjusted number of citations (Figure 2, Panel B). We have added these additional analyses to our manuscript and also included a paragraph summarizing these results.

→ New Figure 3: Differences in AAS (Figure 3a) and citations (Figure 3b) by gender combination of first and last authors, and predicted marginal effects of correlational conversion rate between $\log(\text{AAS} + 1)$ and $\log(\text{citations} + 1)$ by gender of first and last authors (Figure 3c) (lines 189-198)

R3.4. Line 112: The authors use the term “conversion rate” which suggests a causal link between AAS and citations. As far as I’m aware, they have not proven a causal link between AAS and citations. Instead, they have identified an association. I would revise this language accordingly.

→ We have revised our language to emphasize that the conversion rate reflects a correlational and not a causal relationship (lines 175 ff.).

REVIEWERS' COMMENTS:

Reviewer #1 (Remarks to the Author):

All my comments have been appropriately addressed.

Reviewer #2 (Remarks to the Author):

I thank the authors for responding to my comments fully.

Reviewer #3 (Remarks to the Author):

The authors have satisfactorily addressed my comments and have performed the additional analyses which I recommended. I believe that these analyses improve the quality of the manuscript. I have a few final recommendations for improvement.

First, the results section contains not just results, but also elements that should be in the methods section and which should be in the discussion. For example lines 164-167 should be relegated to the methods section, and lines 169-173 are commentary that belong in the discussion. These are two examples, but there are others. Moreover, the methods section includes information that should be moved to the results (lines 291-294). These are some examples, but there are others, and I would recommend that the authors work on ensuring that they have the right content in the right sections of the paper.

Second, the manuscript transitions back and forth between the past tense and the present tense. I would recommend that the authors choose one tense and use it consistently throughout the paper.

Third, the manuscript needs a thorough copy edit.

Re: **COMMSBIO-22-3829-T R2 “Gender Disparities in the Attention to Cardiovascular Research”**

→ **Title changed according to Editor’s suggestion “Gender Disparities in Altmetric Attention Scores for Cardiovascular Research”**

Point-by-Point Response to Reviewers

REVIEWERS' COMMENTS:

Reviewer #1 (Remarks to the Author):

All my comments have been appropriately addressed.

Reviewer #2 (Remarks to the Author):

I thank the authors for responding to my comments fully.

Reviewer #3 (Remarks to the Author):

The authors have satisfactorily addressed my comments and have performed the additional analyses which I recommended. I believe that these analyses improve the quality of the manuscript. I have a few final recommendations for improvement.

First, the results section contains not just results, but also elements that should be in the methods section and which should be in the discussion. For example lines 164-167 should be relegated to the methods section, and lines 169-173 are commentary that belong in the discussion. These are two examples, but there are others. Moreover, the methods section includes information that should be moved to the results (lines 291-294). These are some examples, but there are others, and I would recommend that the authors work on ensuring that they have the right content in the right sections of the paper.

→ *Thank you for pointing out ways to improve order and readability of our manuscript, as noted in the marked up manuscript file, we have followed all the above recommendations and beyond, made sure that the sections are clearly ordered and correctly allotted.*

Second, the manuscript transitions back and forth between the past tense and the present tense. I would recommend that the authors choose one tense and use it consistently throughout the paper.

→ *Thank you for pointing out remaining grammar mistakes, we have now made sure we have eliminated text and grammar mistakes.*

Third, the manuscript needs a thorough copy edit.

→ *We completed a thorough copy edit and updated our document according to the journal’s formatting requirements.*